# MAIS: Memory-Attention for Interactive Segmentation

**Mauricio Orbes-Arteaga** [1]                    HENRY.M.ORBES_ARTEAGA@KCL.AC.UK

**Oeslle Lucena** [1]                                    OESLLE.LUCENA@KCL.AC.UK

**Sebastien Ourselin** [1]                          SEBASTIEN.OURSELIN@KCL.AC.UK

**M. Jorge Cardoso** [1]                            M.JORGE.CARDOSO@KCL.AC.UK

[1] *King's College London, London, UK*

**Editors:** Accepted for publication at MIDL 2025

## Abstract

Interactive medical segmentation reduces annotation effort by refining predictions through user feedback. Vision Transformer (ViT)-based models, such as the Segment Anything Model (SAM), achieve state-of-the-art performance using user clicks and prior masks as prompts. However, existing methods treat interactions as independent events, leading to redundant corrections and limited refinement gains. We address this by introducing **MAIS**, a **M**emory-**A**ttention mechanism for **I**nteractive **S**egmentation that stores past user inputs and segmentation states, enabling temporal context integration. Our approach enhances ViT-based segmentation across diverse imaging modalities, achieving more efficient and accurate refinements.

**Keywords:** Interactive Segmentation, Vision Transformers, Fundation Models, SAM.

## 1. Introduction

Automated segmentation has transformed medical imaging, enabling fast delineation of anatomical structures and pathologies. Deep learning models (Isensee et al., 2021; Wasserthal et al., 2023; Hatamizadeh et al., 2022, 2021) perform well on specialized tasks with large, annotated datasets but struggle with anomalies, where high variability in shape, texture, and location demands robust generalization (Diaz-Pinto et al., 2024). Additionally, their reliance on extensive labeled data makes them costly and labor-intensive in clinical settings.

To address these limitations, interactive segmentation frameworks integrate human feedback—such as corrective clicks or scribbles—to iteratively refine predictions (Du et al., 2023; Diaz-Pinto et al., 2024). Vision Transformer (ViT)-based architectures, such as the Segment Anything Model

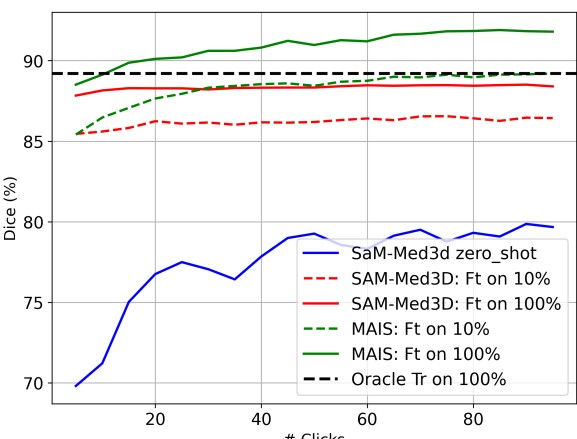

Figure 1: Segmentation accuracy (Dice %) vs. user interactions on **Han-Seg** (Podobnik et al., 2024). SAM-Med3D (3D medical-specific) shows poor zero-shot performance and plateaus after few clicks (fine-tuned), revealing limited task specificity. Our method (MAIS:Ft ), leveraging memory of past interactions, sustains improvement with increasing clicks, approaching Oracle performance even with 10% training data.

(SAM) ([Kirillov et al., 2023](#)), have emerged
as powerful tools for this purpose due to their prompt-driven design and zero-shot capability. Recent efforts, such as fine-tuning SAM on medical images ([Ma et al., 2024](#); [Li et al., 2024](#); [Gong et al., 2023](#); [Wang et al., 2024a](#)), demonstrate promising results. However, a critical bottleneck persists: most implementations process 3D volumes slice-by-slice in 2D, significantly increasing clinician effort during inference ([Mazurowski et al., 2023](#); [Cheng et al., 2023](#)).

Efforts to extend SAM to 3D medical imaging face inherent challenges. For instance, SAM2 ([Ravi et al., 2024](#)), originally designed for video segmentation, treats 3D scans as stacks of 2D slices, akin to video frames. This approach assumes temporal consistency between adjacent slices—a flawed premise in medical imaging, where anatomical cross-sections often exhibit abrupt spatial variations despite representing contiguous structures. Subsequent adaptations like Sam3D ([Bui et al., 2024](#)) attempt to mitigate this by extracting 3D features slice-wise using SAM encoders and aggregating them via lightweight 3D decoders. While this improves efficiency, segmentation quality remains suboptimal, particularly in heterogeneous regions like tumors. Domain-specific architectures such as SAM-Med3D ([Wang et al., 2024b](#)), trained on large-scale medical datasets, enhance zero-shot capabilities but still lack task-specific precision, often requiring excessive user prompts to achieve clinically acceptable results (see Figure [1](#)).

A key limitation of existing approaches is their treatment of user interactions as isolated events. Models like SAM-Med3D handle each correction independently, disregarding past interactions once the mask is updated. This lack of memory leads to redundant corrections, diminishing returns in refinement, and eventual performance plateaus—even with fine-tuning. Our preliminary experiments (see Figure [1](#)) with off-the-shelf SAM-Med3D show that while early interactions enhance segmentation, improvements taper off after a certain point, highlighting the need for models that more effectively incorporate historical context. We hypothesize that incorporating temporal context from past user interactions and segmentation states can overcome these limitations. To this end, we propose a memory-attention mechanism that dynamically stores and retrieves embeddings from sparse user prompts (clicks) and dense segmentation masks. By conditioning predictions on both current inputs and a memory bank of prior interactions, our model enables coherent, incremental refinements across user sessions. Built on top of SAM-Med3D, our method retains the benefits of foundation models—including zero-shot adaptability—while addressing their shortcomings in interactive settings.

The contributions of this work are threefold: (1) A memory-attention mechanism that integrates temporal context into interactive segmentation, enabling robust refinement across user interactions; (2) A lightweight, modular architecture compatible with existing ViT-based frameworks, requiring minimal computational overhead; and (3) Comprehensive validation across multiple modalities and anatomical regions, highlighting the model's adaptability and superiority in low-data scenarios.

## 2. Methodology

### 2.1. Memory-attention for interactive segmentation (MAIS)

Our method is built on SAM-Med3D ViT-based as backbone for 3D interactive segmentation (Wang et al., 2024b). In this framework, a mask decoder processes image embeddings and user prompts to generate 3D segmentation masks. Inspired by SAM2 (Ravi et al., 2024), we introduce a memory attention mechanism that conditions the output on current prompts and also on the memories from past predictions and interactions. Note that in this work, we use the same architecture as SAM-Med3D for the image encoder, prompt encoders, and mask decoder. However, the proposed memory attention mechanism is designed to work with various architectures for these components. Figure 2 illustrates the proposed model, with key components described as follows.

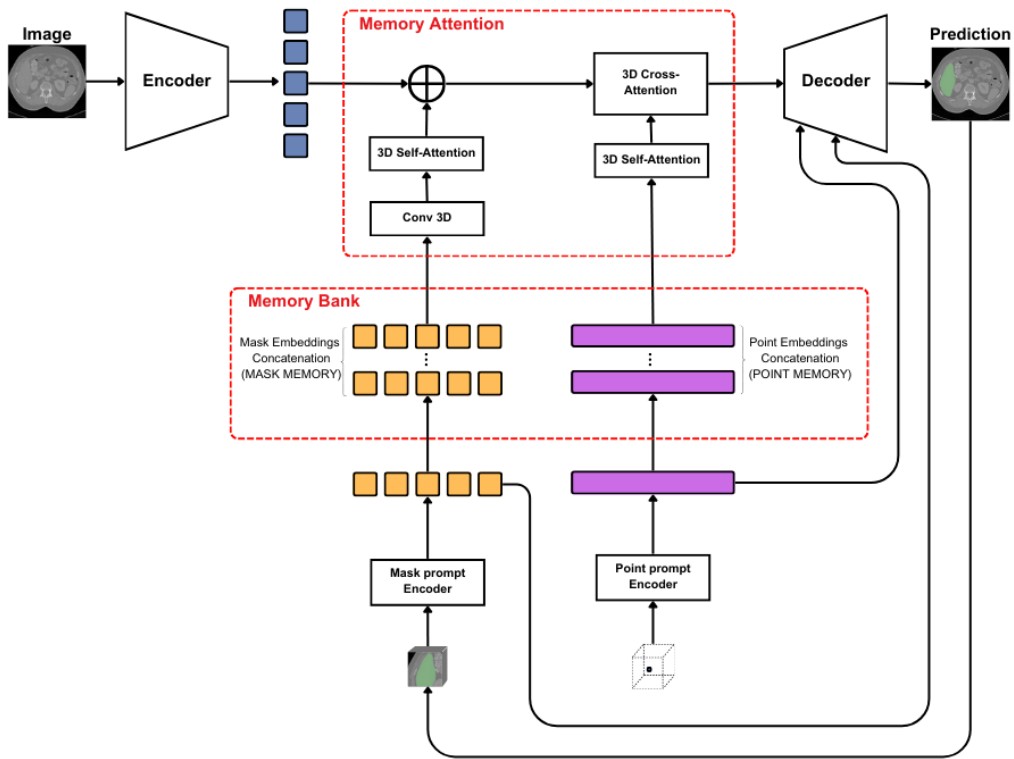

Figure 2: Overview of the proposed segmentation model incorporating memory attention: A 3D image encoder processes the input image generating a 3D embedding. Prompt encoders transform user interactions—positive or negative 3D coordinates—and previous mask predictions into prompt embeddings, which are stored in a memory bank for future interactions. The memory attention mechanism conditions the image embedding on the stored memory before passing it to the mask decoder, which produces the final segmentation output.

### 2.1.1. 3D Image Encoder and Prompt Encoders

The image encoder is a 3D adaptation of the SAM encoder, originally based on ViT. It replaces the 2D convolutional layers with 3D convolutions to process input images and

extends the 2D positional encoding by adding an extra dimension. Additionally, a 3D relative positional encoding (PE) is integrated into the 3D attention blocks. During training and inference, the image encoder processes each image once before prompting the model.

As with most interactive segmentation architectures, we use two types of prompts: **Sparse prompts:** These consist of 3D coordinates derived from point clicks, which can be positive (indicating addition) or negative (indicating deletion). **Dense prompts:** These are 3D masks that represent the current state of the segmentation. Sparse prompts are represented by positional encodings combined with learned embeddings, whereas dense prompts are embedded using convolutional layers.

### 2.1.2. Memory Bank

The memory bank is designed to store information about past predictions and user interactions. After each user interaction, a new memory, composed of sparse and dense embeddings, is created and added to the memory bank. The bank operates as a first-in, first-out (FIFO) queue, retaining the latest $N$ memories. Unlike SAM2, we omit the use of a memory encoder for memory creation. Preliminary experiments show that omitting the memory encoder reduces computational complexity while maintaining performance. The memory bank can be: **Sparse Memory:** this memory bank consists only of click embeddings, meaning that the image embedding is conditioned solely on these sparse prompts through cross-attention, **Dense Memory:** this memory bank consists of only of previous mask embeddings, with the image embedding conditioned on these dense prompts by incorporating the self-attended output of the dense memories, or **Sparse + Dense**: the proposed method (Figure 2), where the memory bank integrates both click and previous mask embeddings. This is the final memory configuration used in MAIS.

### 2.1.3. Memory Attention

The memory attention block conditions the image embedding based on both the memory bank and the current interaction. It first performs self-attention on the sparse and dense memory stacks, generating a dense output, which is then added to the image embedding. Next, a cross-attention is computed between both memory types. To perform self-attention on the dense prompt stack, we employ a Convolutional Transformer block, ensuring computational efficiency while maintaining effective attention processing.

### 2.1.4. Mask Decoder

The mask decoder receives memory-conditioned image embeddings along with the prompt encodings and outputs the segmentation mask. It consists of a stack of "Two-way" Transformer blocks that apply both self-attention and cross-attention to contextualize the prompt tokens with the memory-conditioned image embeddings. (Note that during the first interaction, the image embedding is unconditioned on memory and functions as in the original SAM-Med3D model.) Finally, transposed 3D convolutional layers are used for 3D upscaling, and an MLP is employed at the end to output the segmentation mask.

## 2.2. Datasets

We conducted experiments on publicly available medical imaging datasets, including Computed Tomography (CT) and Magnetic Resonance Imaging (MRI) scans. These datasets contain annotations for multiple anatomical regions, such as the abdomen, head, neck, and heart. Each dataset was split into training (70%) and testing (30%) sets. Notably, none of the images in these datasets were used during the pre-training of SAM-Med3D.

Four medical imaging datasets were used: ACDC (Bernard and Lalande, 2018) (MR, 70 training, 30 testing, 4 heart classes), HaN-Seg (Podobnik et al., 2024) (CT, 28 training, 13 testing, 30 head and neck classes), and the validation set of AMOS (Ji et al., 2022), which we split into AMOS-CT (CT, 70 training, 30 testing, 15 abdomen classes) and AMOS-MR (MR, 14 training, 6 testing, 15 abdomen classes).

## 2.3. Experimental Design

We designed experiments to evaluate the impact of the proposed memory-attention module on interactive segmentation performance and its effect on the few-shot learning potential of foundation models. To investigate this, we adopted a two-stage approach. First, we analyzed the effect of memory bank size and memory embedding types (sparse vs. dense) on segmentation efficacy (Subsection 2.3.1). Second, we conducted extensive external validation across multiple datasets and tasks of varying complexity to assess the robustness and generalizability of our approach (Subsection 2.3.2).

In these experiments, MAIS leverages pretrained weights from SAM-Med3D (Wang et al., 2024b), with the image encoder frozen to preserve foundational feature extraction capabilities. During training, only the prompt encoder and mask decoder are fine-tuned, while the memory attention module is trained from scratch in parallel. we simulate sparse visual prompts by sampling them from regions where previous predictions were incorrect. This approach mirrors human correction behavior, enabling the model to focus on areas requiring refinement.

Training models from scratch on large datasets is out of our scope as it requires substantial resources, making them computationally prohibitive. Furthermore, while they can gain representational capabilities, they may lost specificity for certain applications. Instead, we are interested in improving the adaptive ability of generic interactive models to unseen datasets where data is limited.

### 2.3.1. Memory Bank and Prompt Type Analysis

To investigate key design choices, we conducted experiments on the **HaN-Seg** dataset, focusing on the "Cavity Oral" class. These experiments assessed how memory bank size and prompt types (sparse vs. dense) memory embeddings influence segmentation performance.

**Impact of Memory Bank Size:** We examined whether increasing the memory bank size improves the model's ability to leverage past interactions for better segmentation and how performance scales with additional memory capacity. To this end, we trained **MAIS** with varying memory bank sizes, storing N = 10, 20, 30, 40, 50, 60 and 80 embeddings. The model refined its predictions iteratively using user-provided clicks, accumulating information at each step. Segmentation accuracy was measured using the Dice score (%) at each

iteration. Additionally, we included an oracle baseline (nn-UNet) to establish the upper bound of achievable performance.

**Sparse vs Dense Memory Embeddings** To evaluate the influence of different prompt types in constructing the memory bank, we compared three variations of our method, as described in Section 2.1.2. Specifically, we assessed the impact of using sparse (click-based) memory, dense (mask-based) memory, and their combination. A version of **MAIS** removing the memory attention block (no memory) was included as a baseline for comparison. The models were trained on varying numbers of images (1, 2, 4, and 28) and evaluated using different numbers of user clicks (5, 10, 20, and 50).

### 2.3.2. SEGMENTATION PERFORMANCE

We evaluated segmentation performance across the four datasets (section 2.2) by fine-tuning models with varying amounts of training samples and assessing their performance on validation sets. The following models were compared: **MAIS:** The proposed model (Figure 2) with a memory size of N = 60, using both sparse and dense embeddings. **Ft-SAM3D:** A variant of the model in Figure 2 with the memory attention module removed, effectively reducing it to a fine-tuned version of vanilla **SAM-Med3D**. Similar to the **MAIS** model, only the mask decoder and prompt encoder parameters were updated during fine-tuning. Furthermore, we test two baseline models for performance comparison: **Oracle: nn-UNet** (Isensee et al., 2021) trained from scratch on target data, providing a reference for state-of-the-art performance in task-specific segmentation. **SAM3D (Zero-shot):** The off-the-shelf **SAM-Med3D** model was used to segment the validation sets directly, serving as a benchmark for zero-shot performance of foundation segmentation models on unseen data.

## 3. Results

### 3.1. Memory Bank and Prompt Type Analysis

**Impact of memory bank size** Figure 3-a illustrates the effect of memory size on segmentation performance. The results demonstrate that increasing memory size generally improves performance, with larger memory bank sizes (over 50) approaching/outperforming oracle performance. In contrast, smaller memory sizes (10 and 20) show lower Dice scores, indicating that limited memory restricts the model's ability to refine predictions effectively. The performance gap is most prominent in the early stages (fewer clicks), where larger memory sizes exhibit steeper improvements. However, as the number of clicks increases, the performance differences among larger memory sizes diminish, suggesting a saturation effect. These findings highlight the critical role of the size of the memory bank in achieving accurate and efficient segmentation. It important to note that larger memory sizes require more computational resources due to increased attention computations, making the model more resource-intensive. Thus, a tradeoff between memory size and efficiency is necessary to balance performance and resource usage.

**Sparse vs dense memory embeddings** The results are presented in Figure 3-b. The Sparse Memory approach consistently underperforms compared to Dense Memory, indicating that click embeddings solely are less effective for refining segmentations. The

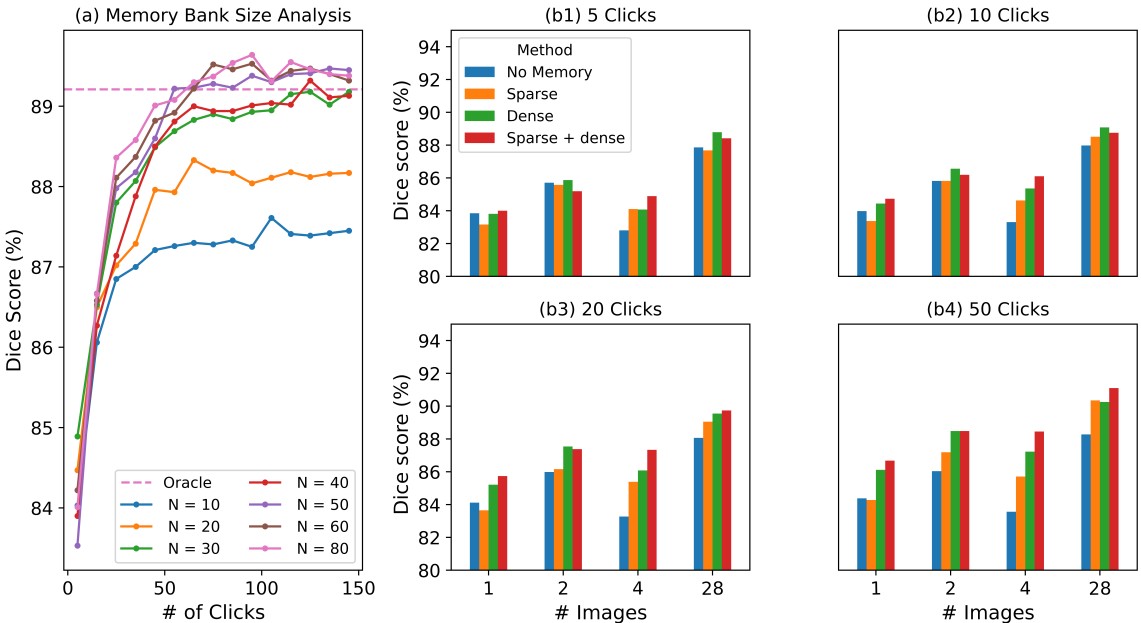

Figure 3: Memory bank and prompt type analysis : Subplot **(a)** Impact of memory bank size on segmentation performance. Dice score (%) is shown as a function of the number of clicks for different memory sizes. Subplots **(b)** Sparse vs dense memory embeddings comparison for the different number of clicks and images used for training.

Sparse + Dense configuration achieves the highest Dice scores, especially at higher click counts (Figure 3-b4, 50 clicks), demonstrating the advantage of combining both memory types. As the number of clicks increases, the performance gap widens, showing that more interactions help the model better utilize stored information. The baseline without memory attention performs worst, highlighting the importance of an effective memory mechanism

### 3.2. Segmentation Performance

Table 1 summarizes the segmentation performance of the evaluated methods across three datasets described in Section 2.2 (See qualitative results in Appendix E and D). The memory bank configuration (Sparse + Dense) was selected based on its superior performance in the previous analysis.

MAIS was found to consistently outperform Ft-SAM3D fine-tuning, particularly as the number of interactions increases. The performance gap is most pronounced in low-data regimes (10% and 50% of the data), where our model achieves significant improvements over Ft-SAM3D, demonstrating its effectiveness when training data is limited. Additionally, in the one-shot scenario, our method significantly outperforms Ft-SAM3D, except on the ACDC dataset. Regarding the impact of interaction count, we observe that increasing user interactions generally improves segmentation performance across all interactive methods. Notably, the performance gap between MAIS and Ft-SAM3D widens as more interactions are allowed, with Ft-SAM3D showing diminishing gains after 10 interactions, whereas our

| # Clicks | One-shot | | | | 10 % of the data | | | 50 % of the data | | | 70 % of the data | | |
|---|---|---|---|---|---|---|---|---|---|---|---|---|---|
| | SAM3D | Ft-SAM3D | MAIS | Oracle | Ft-SAM3D | MAIS | Oracle | Ft-SAM3D | MAIS | Oracle | Ft-SAM3D | MAIS | Oracle |
| **HaN-SEG - CT** | | 1 files | | | 3 files | | | 14 files | | | 28 files | | |
| Auto | - | - | - | 45.65 | - | - | 50.57 | - | - | 54.75 | - | - | 58.22 |
| 1 | 23.26 | 35.51 | **36.61** | - | 38.92 | **39.24** | - | 44.5 | **44.63** | - | 44.56 | **45.57** | - |
| 10 | 34.0 | 40.04 | **42.82** | - | 41.45 | **43.96** | - | 45.46 | **47.55** | - | 45.33 | **49.68** | - |
| 20 | 36.06 | 40.34 | **44.98** | - | 41.56 | **45.86** | - | 45.51 | **49.49** | - | 45.33 | **51.78** | - |
| 50 | 39.07 | 40.56 | **48.51** | - | 41.68 | **48.98** | - | 45.56 | **52.46** | - | 45.43 | **55.7** | - |
| 150 | 41.01 | 40.81 | **51.75** | - | 41.85 | **52.58** | - | 45.59 | **56.4** | - | 45.48 | **59.54** | - |
| **ACDC - MR** | | 1 files | | | 7 files | | | 35 files | | | 70 files | | |
| Auto | - | - | - | 42.93 | - | - | 89.47 | - | - | 92.19 | - | - | 93.25 |
| 1 | 46.94 | **62.65** | 61.38 | - | **73.95** | 70.82 | - | **77.12** | 74.93 | - | **78.73** | 78.02 | - |
| 10 | 68.33 | **72.61** | 70.46 | - | 79.23 | **81.03** | - | 81.36 | **82.98** | - | 82.52 | **84.86** | - |
| 20 | 73.39 | **73.49** | 71.85 | - | 79.84 | **81.72** | - | 81.51 | **84.21** | - | 82.76 | **85.75** | - |
| 50 | 75.98 | **74.66*** | 74.09 | - | 79.84 | **82.53** | - | 81.8 | **85.46** | - | 82.9 | **86.56** | - |
| 150 | 77.75 | 75.22 | **76.54*** | - | 80.21 | **84.27** | - | 81.9 | **86.49** | - | 82.92 | **87.87** | - |
| **AMOS - CT** | | 1 files | | | 7 files | | | 35 files | | | 70 files | | |
| Auto | - | - | - | 19.41 | - | - | 78.12 | - | - | 88.08 | - | - | 89.37 |
| 1 | 79.85 | **77.58*** | 77.14 | - | 78.23 | **78.48*** | - | **78.9*** | 78.89 | - | 79.33 | **79.38*** | - |
| 10 | 83.79 | 80.55 | **82.19*** | - | 81.73 | **82.82*** | - | 82.31 | **83.88** | - | 82.82 | **84.1** | - |
| 20 | 84.29 | 80.97 | **82.92*** | - | 82.15 | **84.34** | - | 82.89 | **85.08** | - | 83.44 | **85.54** | - |
| 50 | 84.77 | 81.35 | **83.24*** | - | 82.32 | **85.41** | - | 83.24 | **86.87** | - | 83.77 | **87.35** | - |
| 150 | 85.09 | 81.68 | **83.5*** | - | 82.58 | **86.24** | - | 83.38 | **88.11** | - | 83.94 | **88.63** | - |
| **AMOS - MR** | | 1 files | | | 2 files | | | 7 files | | | 14 files | | |
| Auto | - | - | - | 29.0 | - | - | 44.75 | - | - | 64.72 | - | - | 85.52 |
| 1 | 74.66 | 70.18 | **71.09*** | - | 72.07 | **72.29*** | - | **73.69*** | 73.65 | - | 73.78 | **74.18*** | - |
| 10 | 79.42 | 75.26 | **78.00*** | - | 75.94 | **78.42*** | - | 78.64 | **79.64** | - | 79.02 | **80.36** | - |
| 20 | 81.15 | 75.75 | **79.48*** | - | 76.46 | **79.57*** | - | 78.91 | **81.1** | - | 79.51 | **81.74** | - |
| 50 | 81.99 | 76.41 | **80.74*** | - | 77.19 | **81.22*** | - | 79.18 | **83.17** | - | 79.63 | **83.78** | - |
| 150 | 82.13 | 76.87 | **81.71*** | - | 77.19 | **82.55** | - | 79.37 | **84.92** | - | 79.85 | **85.85** | - |

Table 1: Comparison of Dice performance for MAIS against Ft-SAM3D, and Baseline methods, Oracle, and SAM3D across multiple datasets when varying amounts of training data for fine-tuning. The best-performing interactive method in each configuration is highlighted in **bold**. Interactive results outperforming the Oracle are underlined, while those underperforming SAM3D are marked with a (*).

method continues improving up to 50 interactions. This suggests that MAIS effectively leverages a large memory capacity to enhance refinement.

Comparing interactive methods against the Oracle nn-UNet, we find that with sufficient training data, the Oracle remains the best performer across all datasets. However, in several cases (e.g., HaN-SEG and AMOS-MR), MAIS outperforms or closely matches the Oracle's performance, particularly when a higher number of interactions is allowed. Conversely, the Oracle struggles when trained on limited data, often underperforming compared to interactive methods. Finally, both Ft-SAM3D and MAIS consistently outperform the zero-shot SAM3D baseline, demonstrating the effectiveness of fine-tuning foundation models for medical image segmentation.

## 4. Conclusions

In this work, we introduced Memory-Attention for Interactive Segmentation (MAIS) to address the limitations of traditional Vision Transformer (ViT)-based approaches in interactive segmentation. By incorporating temporal context through a memory bank that stores past user interactions and segmentation masks, our method significantly enhances segmentation performance across various medical imaging datasets, including MRI and CT scans. The lightweight architecture of the proposed attention module enables effective training

even with limited data, achieving competitive performance against state-of-the-art task-specific models such as nn-UNet. Our experiments demonstrate that increasing memory capacity leads to more effective segmentation refinement as user interactions grow. These capabilities are crucial for developing interactive segmentation tools that enhance clinical workflows, allowing clinicians to improve labeling efficiency and accuracy in medical imaging applications. Future work will integrate the proposed attention mechanism with additional pretrained backbone architectures.

## Acknowledgments

This research was supported by Wellcome/ EPSRC Centre for Medical Engineering (WT203148/Z/16/Z, 213038/Z/18/Z ), Research capability fund (PJ11439), The London AI centre for Value-based Heathcare and GE Healthcare.

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

## Appendix A. Implementation and Computational Details

Our training settings follow those used in (Wang et al., 2024b), with some modifications. We employ different learning rates for different parts of our model: the prompt encoders and mask decoder are optimized with an initial learning rate of 8e-5, while the memory attention parameters use an initial learning rate of 8e-4. Both learning rates follow a multi-step scheduler, decreasing by a factor of 0.1 at epochs 129 and 180. We use the Dice loss function and the AdamW optimizer, training our models for 200 epochs. Input images are cropped into 128×128×128 3D patches centered on a foreground voxel. Training was conducted on an NVIDIA GeForce RTX-4090(24GB).

## Appendix B. Model Complexity MAIS vs SAM2

We compute the number of parameters for each component in the MAIS and SAM2 models to analyze their relative complexity. In MAIS, the memory module consists solely of the memory attention block, whereas SAM2 incorporates an additional memory encoder that processes both image embeddings and output masks. Additionally, we calculate the number of parameters being fine-tuned, which includes both the memory attention module and the mask decoder. Notably, the image encoder is excluded from fine-tuning, as it constitutes the majority of the total model parameters—90% in MAIS and 94% in SAM2—making its optimization computationally impractical. The parameter breakdown in Table 2 highlights the efficiency of the proposed method in terms of model complexity. The memory attention module in our approach (MAIS) comprises only 2.84M parameters, which represents 29.79% of the fine-tuned parameters and merely 2.77% of the total model parameters. In contrast, SAM2's memory module is significantly heavier, with 7.31M parameters, corresponding to 63.42% of the fine-tuned parameters and 3.26% of the total model. This difference underscores the efficiency of our method in maintaining a lightweight memory mechanism while achieving comparable functionality. Additionally, our model does not require a memory encoder, further reducing computational complexity compared to SAM2.

| Component | # Parameters | | % of FT Parameters | | % of total Parameters | |
| --- | --- | --- | --- | --- | --- | --- |
| | MAIS | SAM2 | MAIS | SAM2 | MAIS | SAM2 |
| Memory Attention | 2.84M | 5.92M | 29.79% | 51.40% | 2.77% | 2.64% |
| Memory Encoder | - | 1.38M | - | 12.02% | - | 0.62% |
| **Total Memory module** | 2.84M | 7.31M | 29.79% | 63.42% | 2.77% | 3.26% |
| Mask decoder | 6.69M | 4.22M | 70.21% | 36.58% | 6.53% | 1.88% |
| **Total parameters finetuned** | 9.53M | 11.52M | - | - | 9.30% | 5.13% |
| Image encoder | 92.92M | 212.70M | - | - | 90.69% | 94.77% |
| **Total parameters** | 102.46M | 224.43M | - | - | - | - |

Table 2: Breakdown of total parameters, finetuned parameter share, and overall model parameter distribution for SAM2 and MAIS across different components, highlighting the differences in memory and mask decoder contributions

## Appendix C. MAIS training and inference cost

To evaluate the computational impact of the proposed memory attention mechanism, we measured GPU memory usage and training time required to train and infer MAIS on a single-image, single-label dataset while simulating 150 user interactions. We compared different memory configurations, including No Memory (equivalent to FT-SAM3D), Sparse, Dense, and a combination of Sparse + Dense memory banks.

The results presented in Table 3 demonstrate that the additional computational burden introduced by the proposed memory attention mechanism remains minimal, particularly during inference. During training, GPU memory consumption increases gradually with larger memory sizes but remains within a reasonable range. For instance, in the most demanding configuration (Sparse + Dense with 60 stored embeddings), GPU usage reaches 7164 MB, representing a 32.7% increase over the No Memory baseline (5398 MB), while training time per image rises by approximately 62% (from 14.65 min to 23.75 min). However, for more moderate configurations (e.g., Sparse 10 or Sparse + Dense 10), the increase is minimal, suggesting that efficient memory management can significantly reduce overhead.

Inference results indicate that the added memory mechanism has an even smaller impact on computational cost. The No Memory baseline requires 2756 MB and 27 seconds per image, while the heaviest memory configuration (Sparse + Dense 60) increases memory usage by only 16.5% (3210 MB) and inference time by just 6 seconds (from 27s to 33s). This suggests that, unlike training, where memory size plays a more significant role, inference benefits from the efficiency of the proposed attention module, maintaining near-real-time performance.

In summary, the proposed memory mechanism introduces a modest increase in computational cost during training while remaining nearly imperceptible during inference. Moreover, the parameter analysis confirms that our attention module is significantly more efficient than SAM2, enabling improved memory integration without excessive resource demands. These findings demonstrate that our approach achieves a favorable balance between efficiency and performance, making it well-suited for real-world applications.

| Method conf | Memory Size | Training | | Inference | |
|---|---|---|---|---|---|
| | | GPU Memory (MB) | Time (min) | GPU Memory (MB) | Time (s) |
| No Attention | 0 | 5398 | 14.65 | 2756 | 27 |
| Sparse | 10 | 5578 | 14.82 | 2774 | 28 |
| Sparse | 20 | 5940 | 14.98 | 2846 | 28 |
| Sparse | 60 | 6260 | 16.08 | 3100 | 32 |
| Dense | 10 | 5690 | 15.72 | 2864 | 28 |
| Dense | 20 | 5954 | 17.80 | 2904 | 31 |
| Dense | 60 | 6552 | 21.40 | 3208 | 32 |
| Sparse + Dense | 10 | 5930 | 16.00 | 2836 | 28 |
| Sparse + Dense | 20 | 6370 | 18.55 | 2906 | 29 |
| Sparse + Dense | 60 | 7164 | 23.75 | 3210 | 33 |

Table 3: Comparison of GPU memory usage and computation time for training and inference under different memory configurations.

## Appendix D.  Qualitative results-MR

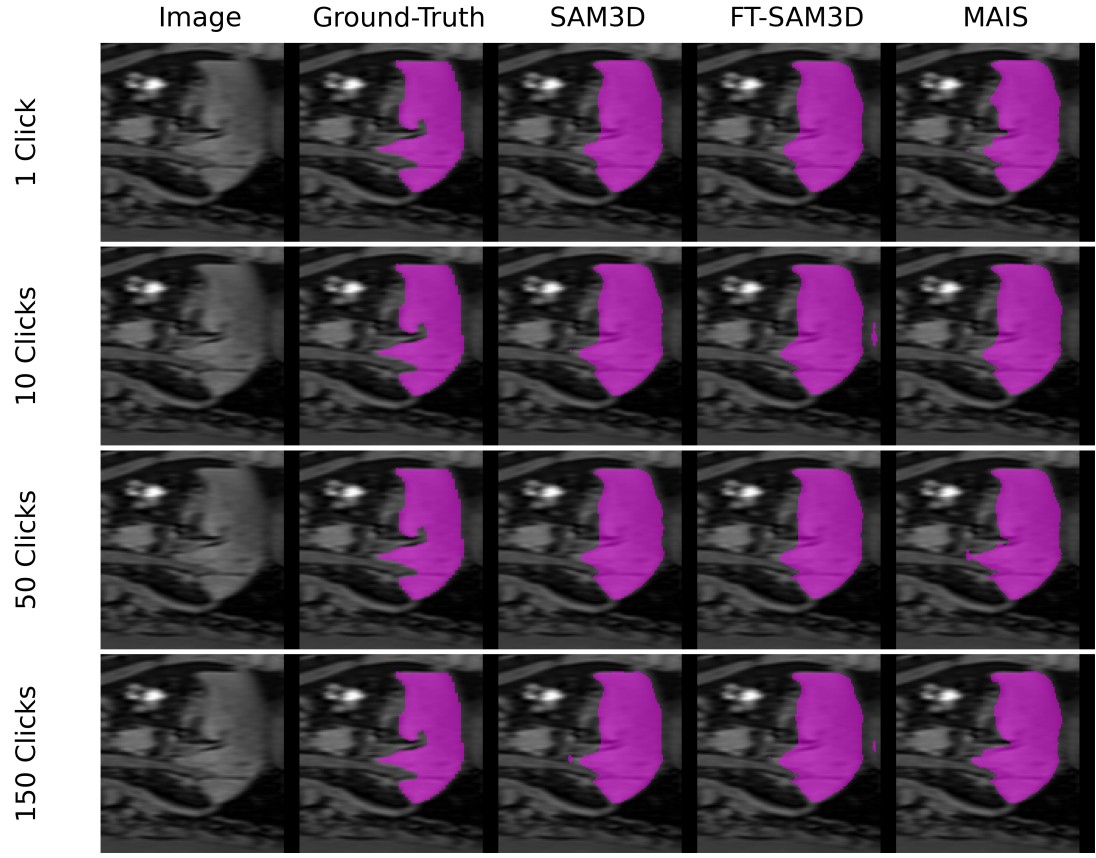

Figure 4: Qualitative result on AMOS-MR dataset For **Liver** with varying numbers of user clicks. The first column shows the original MRI images, and the second column presents the ground-truth segmentations. The remaining columns compare segmentations produced by SAM3D, fine-tuned SAM3D (FT-SAM3D), and MAIS.

## Appendix E. Qualitative results-CT

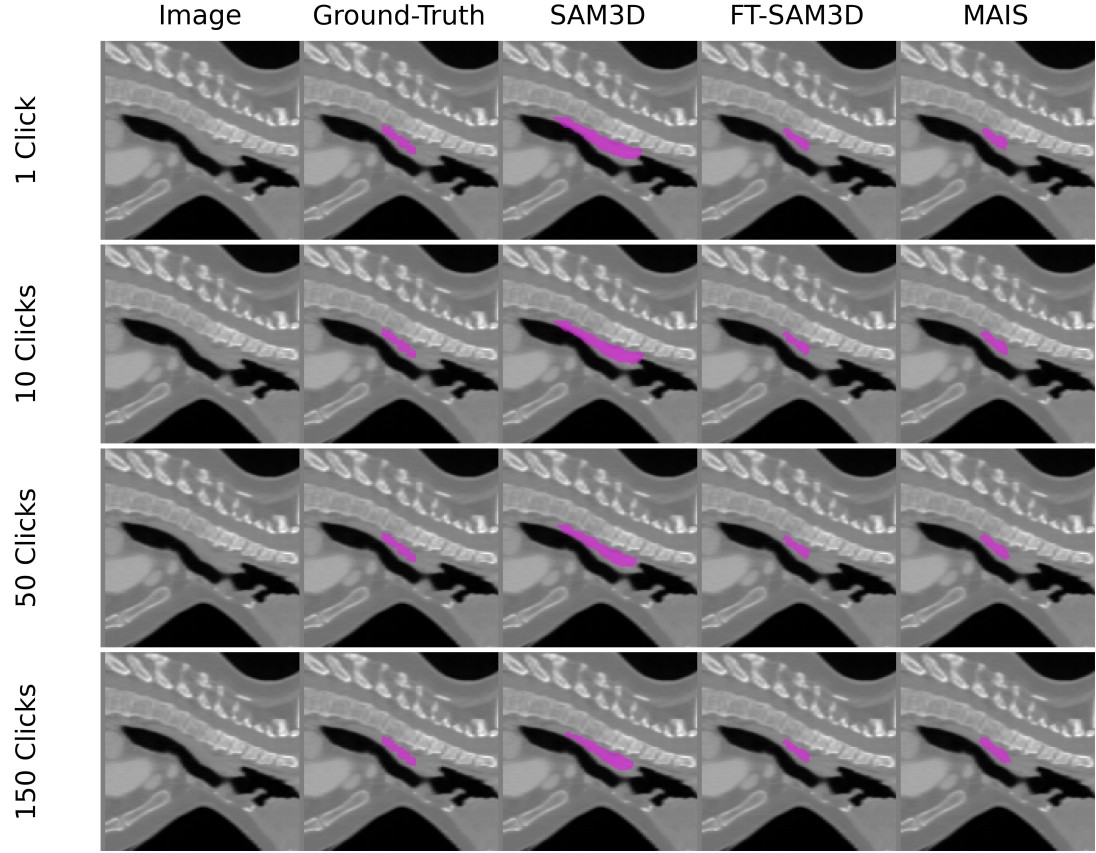

Figure 5: Qualitative result on the HaN-SEG dataset (CT images) for **Esophagus** segmentation with different numbers of user clicks. The first column shows the original CT images, while the second column presents the ground-truth segmentations. The remaining columns compare segmentations produced by SAM3D, fine-tuned SAM3D (FT-SAM3D), and MAIS. It can be observed that SAM3D tends to oversegment the esophagus, while FT-SAM3D undersegments it. MAIS provides more balanced results as the number of clicks increases.

