# OpenReview forum: "MAIS: Memory-Attention for Interactive Segmentation"
_MIDL.io/2025/Conference — MIDL 2025 Poster_

### Official Review · Reviewer_1bPg · 2025-02-15

**Confidence:** 5
**Preliminary Rating:** 2
**Recommendation:** Poster
**Final Rating:** 3

**Summary:**

This paper introduces MAIS, a memory-attention mechanism for interactive 3D medical image segmentation. By storing past user inputs and segmentation states, predictions are refined iteratively. Compared to fine-tuned SAM3D, MAIS improves performance in low-data settings. However, it lacks comparison with MedSAM2, does not clearly differentiate from SAM2, and fails to justify its design choices. Experimental comparisons are unfair, and key efficiency metrics like inference time are missing. Without proper baselines and ablation studies, the claimed advantages remain uncertain.

**Strengths:**

1. Improved Performance in Low-Data Regimes
Experiments show MAIS outperforms fine-tuned SAM3D when data is scarce, demonstrating its ability to leverage historical interactions effectively for segmentation refinement.

2. Computationally Efficient Modular Design
The proposed memory attention mechanism integrates with ViT-based architectures while maintaining efficiency, making it adaptable to existing segmentation models with minimal computational cost.

**Weaknesses:**

1. Lack of Comparison with MedSAM2 and SAM2
The study omits a key comparison with MedSAM2, a strong 3D segmentation model. It also does not clearly explain how MAIS differs from SAM2’s memory mechanism in efficiency or accuracy.

2. Missing Efficiency Metrics
There’s no runtime analysis of inference cost when increasing memory size, which impacts real-world applicability.

3. Unfair Experimental Setup and Lack of Ablation Study
The experiments do not test SAM3D with an optimized memory mechanism for a fair comparison. There is also no ablation study on the impact of memory attention on performance.

4. Unreasonable Setting
The setting proposed in this paper requires multiple interactions to refine the segmentation results after each segmentation. However, since this method modifies the entire 3D volume in each interaction, it means that after each interaction, the entire 3D volume needs to be re-predicted and manually checked. This is not reasonable in practical applications.

**Detailed Comments:**

1.SAM2 vs. MAIS Needs Clearer Distinction
SAM2 already integrates memory for segmentation refinement. How does MAIS improve upon it? The paper should explicitly highlight what makes its memory mechanism more effective.

2. Computational Cost of Memory Attention Unclear
Increasing memory size raises computation, yet the paper claims efficiency. A runtime comparison with SAM3D is needed to confirm that MAIS remains lightweight with larger memory banks.

**Justification Of The Final Rating:**

I appreciate the authors' detailed responses, which have addressed some of my concerns regarding the interactive setting and clinical applicability. The clarifications on interaction timing and the practical workflow are helpful in understanding the system's real-world utility.

However, I remain unconvinced by the justification for omitting comparisons with SAM2  and other variants in 3D interactive segmentation. Beyond MedSAM2, there are numerous approaches adapting foundation models to 3D interactive segmentation that could serve as meaningful baselines. Including such comparisons would significantly strengthen the paper's contribution and place it more clearly within the current landscape of interactive segmentation research.

Additionally, I find the ablation studies on the memory mechanism insufficient. The authors have not adequately addressed my concern about whether low-quality initial memory embeddings might negatively impact segmentation performance. In video object segmentation, low-quality memory often leads to error accumulation - a similar issue could reasonably arise in the proposed 3D memory bank. Figure 3 shows overall performance improvements with increased bank size, but doesn't specifically analyze whether early, potentially inaccurate embeddings might degrade results. More targeted ablation experiments exploring this potential limitation would be valuable.

While I acknowledge the progress made in addressing my initial concerns, these remaining issues regarding comparative evaluations and memory quality analysis still need to be addressed to fully demonstrate the method's advantages and limitations. Therefore, I would change my initial rating from weak reject to borderline.

**Justification Of The Preliminary Rating:**

MAIS is promising but lacks fair comparisons, proper ablation studies, and efficiency analysis. Without clear justification of its memory mechanism and computational cost, its advantages over existing methods remain uncertain.

**Questions To Address In The Rebuttal:**

1. Why is MedSAM2 Missing from Comparisons? MedSAM2 is a strong baseline for 3D medical segmentation. Would MAIS still perform better against it?

2. How Does MAIS Differ from SAM2’s Memory Mechanism? SAM2 also incorporates memory. What unique advantage does MAIS provide over SAM2 in interactive segmentation?

3. What Is the Computational Cost of a Large Memory Bank? How does inference time scale as the number of stored interactions increases? Would it still be practical in real-world use?

4. Justify whether the proposed interactive framework is practical in real-world use.

**Special Issue:**

No

---

> ### Author Response · Authors · 2025-03-08
>
> Q1: Why is MedSAM2 Missing from Comparisons? MedSAM2 is a strong baseline for 3D medical segmentation. Would MAIS still perform better against it?
>
> A1: Our primary goal is to demonstrate how memory attention enhances editing capabilities in ViT-based interactive models. Thus, comparing MAIS against FT-SAM3D (i.e., MAIS without memory) provides the most direct validation of our contribution. Note that we  use FT-SAM3D  as a short name for SAM-Med3D  architecture (not to be confused with SAM3D [1]. (see our comment on review 2 to see why to chose SAM-med3D as the base architecture to test our method)
>
> We acknowledge that MedSAM2  is a strong baseline, but it was primarily optimized for one-click segmentation rather than refinement of segmentations, and most of its evaluations focus on that scenario. In contrast, our work uses nnUNet as a well-established segmentation baseline. As future work, we plan to include MedSAM2, along with other methods for comparison, particularly in low-data fine-tuning settings and refinement scenarios, which are central to our study.
>
>  Note that our proposed memory attention is designed to work with different ViT-based frameworks. Extensive validation across multiple architectures is beyond the scope of this work due to page limitations but remains an important direction for future research.
>
>
> Q2: How Does MAIS Differ from SAM2’s Memory Mechanism? SAM2 also incorporates memory. What unique advantage does MAIS provide over SAM2 in interactive segmentation?
>
>
> A2: the main differences between SAM2 and MAIS attention mechanisms are the following:
>
>  - **Segmentation Objective:** SAM2 is designed for 2D temporal segmentation, where memory is used to retrieve past frames and prompts to assist with segmenting the current frame. This leads to a dynamic, pixel-wise segmentation process—for instance, a pixel labeled as foreground when a person is standing may shift to background when they sit. In contrast, MAIS focuses on static 3D volumetric data, where segmentation objective remains consistent across interactions. Our memory mechanism refines segmentation based only on  user previous inputs (memories), rather than tracking changes over time.
>
>  - **Architectural Differences:** These differing objectives also shape the architecture of the two models. SAM2 stores image embeddings for each frame, requiring a memory encoder to process both the embeddings and output masks. In MAIS, the image embedding is fixed and only previous prompt embeddings are stored, reducing complexity (No memory encoder is needed). Moreover, SAM2 uses more computationally demanding cross-attention and self-attention operations, while our approach is more lightweight (We had added a section that analyse  our model efficiency.)
>
> **Advantage**
>
> MAIS benefits from differences in its segmentation objective, which not only simplifies its architecture but also enables direct processing of 3D inputs through its 3D specialized layers. In contrast, adapting SAM2 for 3D medical imaging requires treating volumetric scans as stacks of 2D slices, akin to video frames. however medical imaging requires spatially consistent treatments for volumetric inputs.
>
>
>  Q3: What Is the Computational Cost of a Large Memory Bank? How does inference time scale as the number of stored interactions increases?
>
> A3: Thanks for raising this question.  We added two sections in the appendix (sup Material) analysing the computational cost and efficiency of the memory attention mechanism. Comparison is done at different sizes for memory bank and different configurations
>
> Our results show that the memory mechanism adds moderate overhead during training but minimal impact on inference. The largest configuration (Sparse + Dense, 60 embeddings) increases GPU memory by **32.7%** and training time by **62%** (14.65 to 23.75 min per image), while moderate setups have negligible impact. At inference, memory usage rises by **16.5%**, adding just **6 seconds**, with the heaviest setup taking 33 seconds (150 interactions).
>
> Additionally, our parameter analysis highlights MAIS’s efficiency over SAM2. Its memory module has **2.84M parameters** (**29.79% of fine-tuned, 2.77% of total**), compared to SAM2’s **7.31M** (**63.42% of fine-tuned, 3.26% of total**).
>
>  Q4: Justify whether the proposed interactive framework is practical in real-world use.
>
> A4: With a 33-second inference time for a single image simulating 150 clicks (less than a second per click), we believe our approach is highly practical for real-world applications, particularly in tools like MONAILabel, which focus in improve the efficiency of annotators' editing workflows (which requires model of models with good editing capabilities) . It's important to note that the most computationally expensive step—image embedding computation—is performed just once, not at each interaction, ensuring that subsequent edits are much faster and do not incur additional computational overhead.
>
> [1] Wang "Sam-med3d:

---

> > ### Comment · Reviewer_1bPg · 2025-03-08
> >
> > I appreciate the authors' providing supplementary experiments to address my concerns. However, I still have the following questions:
> >
> > 1. I would like to clarify whether the 3D segmentation is updated after the user provides a single point or if multiple points are required before updating the 3D segmentation.
> >
> > 2. I'm interested in knowing the time required for a single interaction during the interactive process. Specifically, at time t, after the model receives a point prompt, how much time is needed to generate the segmentation result at time t+1? This interaction time could significantly impact the user experience.
> >
> > 3. Does the updated segmentation result change the entire 3D segmentation output? If so, would users need to inspect the complete 3D segmentation after each interaction? Since each segmentation result could alter the entire 3D volume, there's a possibility that artifacts might be introduced at certain locations during the interaction process.
> >
> > 4. Regarding the comparison with SAM2, why was Hiera-large chosen for the complexity comparison experiment? For my previous experiments, I used Hiera-base+, which has only about 80 MB of parameters, whereas you used a ViT-base backbone with around 100 MB.
> >
> > 5. As a discussion point (no additional experiments required): You mention in the paper that memory stores both dense and sparse embeddings, where I understand dense embeddings to be segmentation results from previous steps. Based on the interactive segmentation results, the initial results are relatively poor, while later results become increasingly accurate. Intuitively, I would think that storing only the more precise dense prompts from later stages might be more beneficial. However, Figure 3 shows that overall performance improves as bank size increases. My question is: Does this improvement primarily come from the point prompts rather than the dense embeddings?

---

> ### Author Response · Authors · 2025-03-11
> **Answer to follow-up questions**
>
> Thanks again to the reviewer for their valuable insights. Please find below our responses to your questions.
>
>
> **A1:** In our approach, the 3D segmentation is updated after every single point interaction. This design choice is driven by the nature of our memory attention mechanism, which captures the direct relationship between each point prompt and its corresponding output through cross-attention. Therefore, our model relies on immediate feedback, where each user click is meant to refine the segmentation incrementally. Delaying updates would undermine this mechanism, as the temporal correspondences between a user's correction and its effect on the segmentation output becomes less direct.
>
> In addition, from a training perspective, processing multiple inputs together rather than one at a time increases the computational burden. Moreover, when several inputs are aggregated before an update, the resulting changes in the segmentation can be unpredictable, making it more challenging for the system to accurately capture the user's intent.
>
> **A2:** Based on our simulated environment, the running time between interaction t and t+1t  is ~0.21 seconds—this estimate excludes any delay introduced by the annotator’s decision-making process. During this interval, the system performs several operations: it encodes new prompts (processing both the clicks and the previous mask), computes the memory attention, and decodes the mask (Generate segmentation).
>
> **A3:** We agree that updating the entire 3D segmentation after each interaction can potentially introduce artifacts. This is a recognized challenge in interactive segmentation systems [3]. While merging strategies—such as those implemented in FocalClick—offer a means to confine updates and reduce such artifacts, we have not integrated these merging algorithms in our current implementation. This decision stems from our focus on leveraging the memory attention mechanism to capture the relationships between user interactions and segmentation outputs. We believe this mechanism is an important addition that can be integrated into any interactive segmentation pipeline to enhance its performance.
>
> Although our method could potentially incorporate merging algorithms, doing so might disrupt the end-to-end coherence and efficiency of our training process. Nevertheless, we acknowledge this as an important direction for future work and appreciate the reviewer’s valuable insight.
>
> **A4:** It is important to note that our primary focus was not on evaluating SAM2 on medical images—indeed, we have not tested SAM2 in that domain, as explained in our paper. Instead, our analysis builds on prior work [1], which identified SAM2:large (200M) as the best-performing variant. Although SAM2:large has a larger parameter count compared to models such as Hiera-base+ (80 MB) or our ViT-base backbone (approximately 100 MB), the performance trends in are clear: SAM2 underperforms MEDSAM, which in turn is outperformed by Sam-Med-3D. Therefore, we believe that comparing against SAM2:large is both fair and meaningful. Moreover, our main emphasis is on illustrating the proportion of the total fine-tuned parameters that are attributed to the memory attention model. —rather than on a direct size-to-size comparison.
>
> However, we are willing to include SAM-Base (80M parameters) in our analysis if it can provide additional insights.
>
> **A5:** Based on the results in **Figure 3b**, our findings suggest that both dense and sparse embeddings contribute significantly to performance. Notably, even when used independently, dense embeddings seem to achieve better performance than sparse embeddings alone. We believe that, although the initial dense embeddings may be less accurate, they still capture important contextual cues from early segmentation steps that our attention mechanism can recover and refine over time.
>
>
> **References**
> [1] Ma, J., Kim, S., Li, F., Baharoon, M., Asakereh, R., Lyu, H., & Wang, B. (2024). Segment anything in medical images and videos: Benchmark and deployment. _arXiv preprint arXiv:2408.03322_.
>
> [2]Bui, N. T., Hoang, D. H., Tran, M. T., Doretto, G., Adjeroh, D., Patel, B., ... & Le, N. (2024, May). Sam3d: Segment anything model in volumetric medical images. In _2024 IEEE International Symposium on Biomedical Imaging (ISBI)_ (pp. 1-4). IEEE.
>
> [3]Chen, X., Zhao, Z., Zhang, Y., Duan, M., Qi, D., & Zhao, H. (2022). Focalclick: Towards practical interactive image segmentation. In _Proceedings of the IEEE/CVF conference on computer vision and pattern recognition_ (pp. 1300-1309).

---

### Official Review · Reviewer_H9SG · 2025-02-21

**Confidence:** 4
**Preliminary Rating:** 4
**Recommendation:** Poster
**Final Rating:** 4

**Summary:**

The authors introduce MAIS (Memory Attention mechanism for Interactive Segmentation). This is an interactive segmentation methods based on Vision Transformers (SAM-Med3D) that leverages past interactions with the user through a memory bank and attention to explicitly integrate past context. The authors focus on 2 style of prompts: click interactions and segmentation masks. They test on datasets from various modalities (MRI and CT) and on various anatomies.

**Strengths:**

*Intro and related work*
- Clear motivation

*Experiments*
- The authors test on an extensive set of anatomies and  on two modalities.
- The authors also evaluate the impact of different design choices such as the size of the memory bank, and the use of each embedding type (sparse, dense, sparse+dense).
- The authors also evaluate performances in different training regimes (one-shot, 10%, 50% and 70%).


*Visualization*
- Figure 2 is very clear and helps understand the architecture of the model in more details.

*Style*
- The paper is generally well written and quite detailed about their strategy.

**Weaknesses:**

*Baselines*
- Why compare to the SAM3D model *without* memory bank ? Since the memory bank is the main contribution of this paper, it seems that this baseline is critical to evaluate the validity of this method. Maybe I am missing something? This seems like the most important difference with SAM3D models?
- Why not compare with the SAM 2 model trained with the same procedure ? If I understand correctly the memory attention design is similar too (but less efficient?). It would make the results even more impactful.

*Results*
- It would be nice to see performances on other metrics such as surface distance or intersection over union.
- Standard Deviation or boot strapping the results over various split would help give an insight of the variability and significance of the results for all graphs and tables.

* Visualization*
- I would love to see visualizations for more than one dataset and one visualization

**Detailed Comments:**

*Methods*
- The method section is a bit lacking in terms of math. I think it would improve the readers' understanding and make the paper even more compelling to understand clearly the model the authors are using.

*Visualization*
- Figure 1, it would be nice to overlap a grid to map curves to the corresponding Dice score.
- It would also be nice to have more than one visualization per modality to understand MAIS's behavior visually.

**Justification Of The Final Rating:**

I thank the authors for their comments that comfort my confidence in the score I gave. I still think that SAM 2 is a *major* baseline to include, *especially* since the authors thinks it would not be quite as good as expected. This is the main reason why I did not increase my score. Otherwise, I think that this paper is quite clear and would contribute to the community.

**Justification Of The Preliminary Rating:**

The authors introduce a new mechanism to integrate past interactions to inform model prediction and demonstrate it on various dataset. The authors demonstrate MAIS through various training regime experiments and test various design choices such as the size of the memory bank and the embeddings. My biggest concern is, I don't understand why they don't compare SAM3D with the memory bank and with SAM 2. I would also like to see some evidence that the author's model is lightweight as it is part of the main contribution (or remove that contribution).

**Questions To Address In The Rebuttal:**

- Can the author explain why they compare to the SAM3D model *without* memory bank ? Since the memory bank is the main contribution of this paper, it seems that this baseline is critical to evaluate the validity of this method.
- Since efficiency is a core argument (main contribution number 2), could the authors comments on the impact of using  the memory bank in terms of efficiency (parameters, impact at inference and or at training ) ? Especially in the context of Sparse, Dense, (Sparse + Dense) and how they compare with SAM2's framework.
- Why not compare with the SAM 2 model trained with the same procedure ? If I understand correctly the memory attention design is similar too (but less efficient?). It would make the results even more impactful.
- I am also curious about the number of clicks the number of clicks. Is a user using 50 clicks realistic in a 3D interactive segmentation scenario? What about a 150 ? Could the author comments on that ?

---

> ### Author Response · Authors · 2025-03-08
>
> Q1:  Can the author explain why they compare to the SAM3D model _without_ memory bank ? Since the memory bank is the main contribution of this paper, it seems that this baseline is critical to evaluate the validity of this method.
>
> A1: Thanks for rise this question. Our primary goal is to demonstrate how memory attention enhances editing capabilities in ViT-based interactive models. Thus, comparing MAIS against FT-SAM3D (i.e., MAIS without memory) provides the most direct validation of our contribution. Note that we  use FT-SAM3D  as a short name for SAM-Med3D  architecture.
>
>  We acknowledge that FT-SAM3D might be confused with SAM3D [2], a method that processes 3D volumes in a 2D slice-by-slice fashion using traditional SAM but that  yields suboptimal segmentation quality for volumetric data (see introduction).
>
> In contrast, we chose  SAM-Med3D as the base architecture, because it represents the natural transition to a memory-aware 3D architecture, being the 3D adaptation of SAM—just as SAM2 introduced temporal memory. Unlike most SAM adaptations, SAM-Med3D was trained from scratch on 3D medical images, ensuring better alignment with volumetric data. Additionally, it has demonstrated superior performance compared to many 2D adaptations, such as Med-SAM2D.
>
> Finally, while our proposed memory attention mechanism is designed to be compatible with different ViT-based frameworks, extensive validation across multiple architectures is beyond the scope of this work due to page limitations.
>
>
>
> Q2: Since efficiency is a core argument (main contribution number 2), could the authors comments on the impact of using the memory bank in terms of efficiency (parameters, impact at inference and or at training ) ? Especially in the context of Sparse, Dense, (Sparse + Dense) and how they compare with SAM2's framework.
>
> A2: to cope with this concern, we have added two sections in the appendix B and C (see supporting material) analysing the computational cost and efficiency of the memory attention mechanism across different configurations (no memory, sparse, dense, and sparse + dense)
>
> Our results show that the memory mechanism adds moderate overhead during training but minimal impact on inference. The largest configuration (Sparse + Dense, 60 embeddings) increases GPU memory by **32.7%** and training time by **62%** (14.65 to 23.75 min per image), while moderate setups have negligible impact. At inference, memory usage rises by **16.5%**, adding just **6 seconds**, with the heaviest setup taking 33 seconds (150 interactions).
>
> Additionally, our parameter analysis highlights MAIS’s efficiency over SAM2. Its memory module has **2.84M parameters** (**29.79% of fine-tuned, 2.77% of total**), compared to SAM2’s **7.31M** (**63.42% of fine-tuned, 3.26% of total**).
>
>
> Q3: Why not compare with the SAM 2 model trained with the same procedure ? If I understand correctly the memory attention design is similar too (but less efficient?). It would make the results even more impactful.
>
> A3: We appreciate this suggestion; however, these models are not directly comparable due to fundamental differences in their design and objectives (see comments on reviewer 3 for more details). Adapting SAM2 to 3D medical images would require processing volumes slice by slice, fundamentally altering its memory mechanism’s purpose. Unlike our model, which maintains a global volumetric context, SAM2’s memory would operate on sequential slices, limiting its effectiveness in volumetric segmentation.
>
>  Q4: I am also curious about the number of clicks the number of clicks. Is a user using 50 clicks realistic in a 3D interactive segmentation scenario? What about a 150 ? Could the author comments on that ?
>
> A4: As we have mentioned, our work aims to address redundant corrections and diminishing returns in ViT-based interactive segmentation methods. Our primary focus is on applications like **MONAI Label**, where refinement capability is crucial for efficient annotation.
>
> We evaluate up to **150 clicks** to demonstrate that our method continues improving segmentation quality at higher interaction levels, unlike models without memory, which plateau more quickly. However, in practical **MONAI Label** applications, **50 clicks is a reasonable expectation**, particularly during the initial stages of segmenting a dataset.
>
> We acknowledge the existence of architectures  optimized for **one-click segmentation**, such a Medical sam 2 , but these methods often struggle to generalize to unseen tasks, where their ability to refine segmentations remains uncertain. As future work, we plan to include such approaches in our benchmarks for a more comprehensive comparison.
>
>
> References:
>
> [1] Wang, Haoyu, et al. "Sam-med3d: (2023).
>
> [2] Bui, Nhat-Tan, et al. "Sam3d: Segment anything model in volumetric medical images.", 2024.
>
> [3] Zhu,  (2024). Medical sam 2

---

> > ### Comment · Reviewer_H9SG · 2025-03-13
> > **Follow-up questions**
> >
> > Thank you very much for the thorough reply.
> >
> > Q1. Thanks for the clarification.
> >
> > Q3. I am not sure I follow why this makes SAM2 an unacceptable baseline. It seems that it would highlight the need for a global context more clearly. "SAM"-type techniques. What I had in mind when I asked this question was to see the comparison with SAM 2. Especially since the authors mention "Inspired by SAM2, we
> > introduce a memory attention mechanism that [...]". If the authors think that the baseline would do badly, then it would be great to have numerical results to validate this. The fact that SAM 2 interprets it as a temporal dependency will only make MAIS look stronger.
> >
> > Q4. Do the authors know after how many clicks do baseline models without memory typically plateau ?

---

> ### Author Response · Authors · 2025-03-13
> **Answer to follow-up questions**
>
> Thanks again for the insightful questions, we appreciate the opportunity to clarify our rationale.
>
> **A3:**  Previous evaluations of SAM2 in the medical domain have highlighted certain limitations [1,2]. For instance, [1] indicates that SAM2 does not consistently outperform its predecessor, SAM. Specifically, while SAM2 may occasionally surpass SAM in high-contrast modalities such as MR images, its performance in modalities like CT is generally lower. Moreover, [1] shows that even the largest SAM2 model underperforms MEDSAM, which in turn is outperformed by Sam-Med3D [3].
>
> Based on these findings, we focused our comparisons on models that have demonstrated robust performance in medical contexts. We selected nnUNet to represent task-specific models and Sam-Med3D as a benchmark for interactive methods. This approach ensures that our evaluations are grounded in models with established efficacy in medical image segmentation.
>
> Furthermore, we want to reiterate that while our memory proposal is inspired by SAM2, the underlying concept of memory is fundamentally different. SAM2 is designed for 2D temporal segmentation—its memory module retrieves information from previous slices to segment the current one. In contrast, our memory attention mechanism in MAIS is primarily focused on incorporating information from previous user edits. To make it more clear, the only advantage of SAM2 over SAM in the medical imaging domain is its ability to leverage temporal information from previous slices by treating sequential 2D slices as video frames. This approach allows SAM2 to propagate segmentation information across slices, reducing the need for manual edits on each individual slice. Conversely, SAM requires manual editing for each slice independently, as it lacks this temporal context integration.
>
> MAIS is designed for 3D backbones, meaning the complete segmentation is generated at each interaction rather than relying on propagation across slices. Consequently, our memory mechanism is primarily focused on modeling and refining user edits.
>
>
>
> **A4:**  Based on Table 1, we observed that for Ft-SAM3D the performance gains tend to plateau between 10 and 20 interactions across most configurations. In other words, beyond roughly 10–20 clicks, additional user interactions yield diminishing improvements.
>
> **References:**
>
>
> [1]Sengupta, S., Chakrabarty, S., & Soni, R. (2024). Is SAM 2 Better than SAM in Medical Image Segmentation?. _arXiv preprint arXiv:2408.04212_.
>
> [2] Ma, J., Kim, S., Li, F., Baharoon, M., Asakereh, R., Lyu, H., & Wang, B. (2024). Segment anything in medical images and videos: Benchmark and deployment. _arXiv preprint arXiv:2408.03322_.
>
> [3] Wang, Haoyu, et al. "Sam-med3d: (2023).

---

### Official Review · Reviewer_151e · 2025-02-21

**Confidence:** 2
**Preliminary Rating:** 4

**Summary:**

The paper focuses on the interactive segmentation task in the medical domain. It extends MedSAM-3D by incorporating the past interactions as a memory to enhance the segmentation. A novel memory attention and memory bank mechanism is proposed to selectively utilize the past memories. The experiments on MRI and CT datasets show the method's superiority over existing methods.

**Strengths:**

A novel memory mechanism is proposed to enhance the segmentation by utilizing past interactions. The module is universal and can be adapted to different architectures.

The experiments are comprehensive and the ablations studies demonstrate how the memory bank affect the propose model's performance.

**Weaknesses:**

The format of the paper may be further refined; some images are wrongly referred in the text

It would be interesting to see what exactly were stored in the memory bank, and how the memory attention worked on the past interactions

**Detailed Comments:**

The format of the paper may be further refined; some images are wrongly referred in the text

It would be interesting to see what exactly were stored in the memory bank, and how the memory attention worked on the past interactions

**Justification Of The Preliminary Rating:**

The paper proposed a novel memory mechanism to enhance existing SAM models' interactive segmentation ability. The experiments are comprehensive and fully demonstrate the model's performance. Some minor formatting problems could addressed.

**Questions To Address In The Rebuttal:**

what extra computations and memory are brought by the memory bank?

---

> ### Author Response · Authors · 2025-03-08
>
> Q1: what extra computations and memory are brought by the memory bank?
>
> A1: The memory bank introduces additional computations and memory usage through the following steps:
>
> - **Dense Embeddings Processing:** The dense embeddings first pass through a 3D convolutional layer (Conv3D), which downsamples them. This is followed by a self-attention layer (one transformer block) applied to the downsampled embeddings.
>
> - **Self-Attention for Dense Embeddings:** The dense embeddings then undergo another self-attention layer (one transformer block), refining the representation further.
>
> - **Integration with Image Embedding:** The attended dense embeddings are added to the image embedding, and a cross-attention mechanism (two-way transformer block) is applied between the attended encoder and the image embedding, resulting in the attended image embedding (as illustrated in Figure 2).
>
>
> In addition We have added two sections in the appendix (B and C) analysing the computational cost and efficiency of the memory attention mechanism (see supporting material). Comparison is done at different sizes for memory bank and different configurations
>
> Our results indicate that the proposed memory mechanism introduces a moderate computational overhead during training, with minimal impact on inference. The heaviest memory configuration (Sparse + Dense, 60 embeddings) leads to a **32.7% increase in GPU memory usage** and a **62% increase in training time** (from 14.65 to 23.75 minutes per image). However, more moderate configurations result in negligible overhead. At inference, the impact is even smaller, with a **16.5% increase in memory usage** and an additional **6 seconds** required, with the most demanding configuration taking 33 seconds (simulating 150 interactions).
>
> In addition  wrong reference figures have been corrected and improved.

---

### Author Rebuttal · Authors · 2025-03-08

**Rebuttal:**

We thank the reviewers for their thoughtful feedback and are pleased that all recognize the promise of our memory attention mechanism. Below, we address key points raised.

**Baseline Comparisons: SAM3D, SAM2, MedSAM2:**

 Our goal is to show how memory attention enhances ViT-based interactive models editing capabilities. Thus, comparing MAIS against FT-SAM3D (i.e., MAIS without memory) provides the most direct validation of our contribution.

Note that FT-SAM3D refers to the SAM-Med3D architecture (Sec. 2.3.2), not SAM3D [1], a method processes 3D images as a stack of 2D slices and has shown suboptimal segmentation in volumetric data.

Regarding SAM2, it focuses on 2D temporal segmentation, where memory objectives and memory meaning differ from MAIS. Comparing it to MAIS, designed specifically for 3D volumetric segmentation, or using SAM2 as base architecture, would not be meaningful.

MedSAM2 is a strong baseline but is optimized for one-click segmentation rather than refinement. Instead, we use nnUNet as a well-established segmentation baseline. However, we plan to include MedSAM2 along others in future comparisons, particularly in low-data fine-tuning and refinement scenarios.

Finally, our choice of SAM-Med3D over other architectures is detailed in individual responses. MAIS is designed to work with any ViT promptable architecture, and future work will validate it on additional models.

 **MAIS Efficiency and Comparison to SAM2 Memory:**

We agree that memory bank efficiency is crucial. Appendix updates include detailed analyses of computational overhead during training and inference. Our results show moderate training overhead but minimal impact on inference.  Additionally, our memory module is significantly lighter than SAM2’s, with **2.84M parameters** (29.79% of fine-tuned parameters) compared to **7.31M** (63.42% in SAM2), reinforcing its efficiency.

**Real-World Applicability:**

 Our analysis shows that processing a single image with **150 clicks** takes **33 seconds**, averaging **less than a second per click**, demonstrating the practicality of our approach for real-world applications. This is particularly relevant for tools like **MONAILabel**, where efficient annotation workflows require models with strong editing capabilities.

Minor concerns and additional details have been addressed in the individual reviewer responses.

An updated version of the document (Appendix update) has been upload as supporting material.

[1] Bui, "SAM3D

**Supporting Material:**

/attachment/4e4585c02d89b84ecf80d52a96d851e8f56602af.pdf

---

### Author Response · Authors · 2025-03-14
**Possible display Issue on OpenReview: Clarification on Follow-Up Responses**

I would like to thank the reviewers again for their valuable feedback and for this engaging discussion period. I want to clarify that I have responded to both your initial questions and the subsequent follow-up questions promptly and using the official comment feature on OpenReview. However, I’ve noticed that the interface displays all responses in a similar format, which might make it seem as though only the initial questions were addressed.

Please note that the follow-up responses have been provided separately, and notification emails with these responses have been sent to all reviewers. I want to kindly ask the meta-reviewer  to consider that the uniform display format on the platform might not fully capture the distinct and timely nature of my responses.

---

> ### Comment · Reviewer_1bPg · 2025-03-15
> **Response to author clarification**
>
> I agree that the UI is not user-friendly. However, I have carefully reviewed [all] the authors' responses before my final rating, including their replies to other reviewers. I would like to reiterate my concerns:
>
> 1. The lack of comparison with SAM2-based methods adapted to 3D contexts remains a significant limitation. This includes not only MedSAM2 but also simpler approaches that adapt SAM2 2D video functionality to 3D medical imaging. One such straightforward implementation is presented in [1]. The authors have only compared efficiency metrics rather than effectiveness, which makes it difficult for readers to be convinced of this paper's contributions. Including these comparisons would substantially strengthen the manuscript. Even just including the simple comparison results in effectiveness (not only the efficiency) would make the paper significantly more convincing.
>
> 2. Regarding my suggestion that "storing only the more precise dense prompts from later stages might be more beneficial" - while the authors provided an explanation, I remain unconvinced. I believe a more thorough ablation analysis of the memory design would benefit their research. However, I should clarify that this particular point is not a determining factor in my evaluation score.
>
> [1] Shen, C., Li, W., Shi, Y., & Wang, X. (2024). Interactive 3d medical image segmentation with sam 2. arXiv preprint arXiv:2408.02635.

---

> > ### Author Response · Authors · 2025-03-15
> > **Response to reviewer 1bPg final feedback.**
> >
> > We appreciate the reviewer's valuable feedback once again.
> >
> > We would like to emphasize that our proposal is straightforward: **By leveraging past user interactions, interactive segmentation models can achieve significant improvements.**
> >
> > As mentioned in the paper as well as in previous answers to reviewers, we recognise the value of implementing our memory attention mechanism in different architectures, even such as SAM2 and MedSAM2, as well as incorporating them as additional baselines (similar to what we did with SAM-Med3D in this work). However, we defend our decision not to include these alternatives in the current work, based on the reasons previously mentioned. Nonetheless, we consider it appropriate to explore these suggestions for an extended version of the current work.
> >
> > Regarding the inclusion of additional efficiency and effectiveness comparisons, we would like to point out that our evaluation framework—which includes the Dice coefficient, computational time, and GPU consumption (added in the appendix)—aligns with the standards observed in many conference and journal papers in our field, especially in interactive segmentation [1,2,3,4,5,6,7,...].
> >
> > [1] Shen, C., Li, W., Shi, Y., & Wang, X. (2024). Interactive 3d medical image segmentation with sam 2. arXiv preprint arXiv:2408.02635.
> >
> > [2]Wang, H., Guo, S., Ye, J., Deng, Z., Cheng, J., Li, T., ... & Qiao, Y. (2023). Sam-med3d: towards general-purpose segmentation models for volumetric medical images. _arXiv preprint arXiv:2310.15161_.
> >
> > [3] He, Y., Guo, P., Tang, Y., Myronenko, A., Nath, V., Xu, Z., ... & Li, W. (2024). Vista3d: Versatile imaging segmentation and annotation model for 3d computed tomography. _arXiv preprint arXiv:2406.05285_.
> >
> > [4] Bui, N. T., Hoang, D. H., Tran, M. T., Doretto, G., Adjeroh, D., Patel, B., ... & Le, N. (2024, May). Sam3d: Segment anything model in volumetric medical images. In _2024 IEEE International Symposium on Biomedical Imaging (ISBI)_ (pp. 1-4). IEEE.
> >
> > [5]Zhu, J., Hamdi, A., Qi, Y., Jin, Y., & Wu, J. (2024). Medical sam 2: Segment medical images as video via segment anything model 2. _arXiv preprint arXiv:2408.00874_.|
> >
> > [6]Sengupta, S., Chakrabarty, S., & Soni, R. (2024). Is SAM 2 Better than SAM in Medical Image Segmentation?. _arXiv preprint arXiv:2408.04212_.
> >
> > [7] Ma, J., Kim, S., Li, F., Baharoon, M., Asakereh, R., Lyu, H., & Wang, B. (2024). Segment anything in medical images and videos: Benchmark and deployment. _arXiv preprint arXiv:2408.03322_.

---

> > > ### Author Response · Authors · 2025-03-15
> > >
> > > We have already included a reference in our manuscript that demonstrates the underperformance of SAM2 in the medical domain.

---

### Comment · Program_Chairs · 2025-03-17
**Forwarding a message from the authors to AC**

Dear AC,

Please find below a message which we are forwarding from the authors. Could you please make sure to consider all of the author's comments? Thank you.

Forwarded message:

I have a concern regarding how open review is carrying out the discussion period for MIDL.

I initially responded to the reviewers' questions using the "Official Comment" button. When they posted follow-up questions, I also replied using the same method. However, when I view my responses, they appear as if they were made in response to the initial set of questions rather than the follow-ups.

I'm concerned that the meta-reviewers might not realize I have addressed the follow-up questions and may think I haven't responded to them.

---

### Meta-Review · Area_Chair_qPmF · 2025-03-21

**Recommendation:** Accept (Poster)
**Confidence:** 5

**Metareview:**

- Overall Evaluation

This work proposed MAIS, an interactive segmentation framework for medical image analysis. The main idea is to reduce the number of clicks and improve the interactions/consistency between user inputs via the temporal attention memory. Experiments on multiple datasets demonstrated the effectiveness of the proposed method.

- Strength

Interactive segmentation is a popular topic in medical image community since it is important to include the confirmation/refinement from the experts, making the result reliable. The idea of this work is also promising. I can clearly follow the pipeline based on good presentation and solid evaluations in the paper. Considering the comments from the reviewers, I agree with the author's claims that the additional comparisons with SAM2 are not fully emerged since previous literature has demonstrated the issues from SAM2 on medical data. To my understanding, it expected to see more results on 2D data as well as the comparisons with SAMMed-2D.

- Weakness

I am always considering the fair evaluation metrics for interactive segmentation tasks. As we all know that the annotations from human can be quite various even for experted clinicians. Therefore, it is expected to see more user studies to support the results/claims if the authors are planning to extend this work to a journal version. However, this will not affect my positive recommendation of this work.